# Effect of Ageing on Self-Healing Properties of Asphalt Concrete Containing Calcium Alginate/Attapulgite Composite Capsules

**DOI:** 10.3390/ma15041414

**Published:** 2022-02-14

**Authors:** Xin Yu, Quantao Liu, Pei Wan, Jiangkai Song, Huan Wang, Feiyang Zhao, Yafei Wang, Jinyi Wu

**Affiliations:** State Key Laboratory of Silicate Materials for Architectures, Wuhan University of Technology, Wuhan 430070, China; yu-xin@whut.edu.cn (X.Y.); songjk@whut.edu.cn (J.S.); 303561@whut.edu.cn (H.W.); zhaofeiyang@whut.edu.cn (F.Z.); wangyafei@whut.edu.cn (Y.W.); jinyi_wu@whut.edu.cn (J.W.)

**Keywords:** asphalt mixture, self-healing, calcium alginate capsules, ageing treatment

## Abstract

Calcium alginate capsules within asphalt concrete can gradually release interior asphalt rejuvenator under cyclic loading to repair micro cracks and rejuvenate aged asphalt in-situ. However, asphalt pavement will become aged due to environmental and traffic factors during the service period. In view of this, this paper investigated the effect of ageing on the healing properties of asphalt concrete containing calcium alginate/attapulgite composite capsules under cyclic loading. The capsules were fabricated using the orifice-bath method and the morphological structure, mechanical strength, thermal stability, oil release ratios and healing levels of capsules in fresh, short-term ageing and long-term ageing asphalt concrete were explored. The results indicated that the different ageing treatments would not damage the multi-chamber structure nor decrease the mechanical strength of capsules but would induce the capsules release oil prematurely. The premature oil released from capsules in turn can offset the ageing effect owing to ageing treatment. The short-term ageing and long-term ageing plain asphalt mixtures gained strength recovery ratios of 39.3% and 34.2% after 64,000 cycles of compression loading, while the strength recovery ratios of short-term ageing and long-term ageing asphalt mixtures containing capsules were 63.5% and 54.8%, respectively.

## 1. Introduction

Asphalt mixture has been the principal surface paving material on highway due to its outstanding performance. However, after service for years, asphalt pavement would become damaged under repeated loading [1] and environmental factors such as low temperatures [2] and UV exposure [3,4]. Micro cracks will fatally propagate into macro cracks without appropriate maintenance applied to asphalt pavement, hence not only reducing the service life of pavement, but also affecting transportation capacity of the road and the vehicle safety. Thus, to maintain asphalt pavements in optimum condition during their lifespan, external maintenance is usually conducted on pavement by the road agencies The existing maintenance measures focusing on repairing the cracks are all passive methods since they were taken only after macro cracks occur on the road surface. These measures not only consume masses of natural resources but also result in heavy ecological pollution such as greenhouse gas emissions [5,6] and volatile organic compounds emissions [7]. Consequently, intelligent and clean maintenance technologies are highly desirable for asphalt highways.

It is widely known that asphalt is a viscoelastic material that can spontaneously heal the interior micro cracks during resting periods or at high temperature [8,9,10]. However, the self-healing process is quite slow owing to the short intermission during the real service condition. Moreover, it is more difficult for asphalt molecules to flow under the conditions of low temperature and asphalt ageing. Thus, micro cracks cannot be repaired in a prompt and effective manner by themselves. In response to this issue, domestic and global researchers have proposed a novel self-healing method called encapsulated self-healing technology. This is a regenerative encapsulated self-healing technology based on material replenishment. The encapsulation technology which has the characteristics of encapsulation, permeability and stress-controlled release was applied to wrap healing agent in the shell material [11,12,13]. The healing agent (asphalt rejuvenator) will be released under the vehicle loading to heal the micro cracks.

As proven in practice, the rejuvenator in the capsules released into the mixtures not only effectively improve the ability of the asphalt to repair micro-cracks, but also rejuvenate aged asphalt binder in situ due to the substitution of the lost light components [11,12]. Therefore, it is a prospective maintenance technology for future asphalt pavement. The capsules manufactured in this field have multi-chambers wrapping the rejuvenator [14,15,16,17,18]. In contrast to the conventional core-shell type capsules with a one-time fracture release manner [19,20,21], the resilient multi-cavity self-healing capsules (1–10 mm) can progressively release the healing agents in the chambers due to elastic shrinkage without rupture under cyclic loading [22,23]. This type of capsules has the advantages of large coating volume, high healing efficiency, long-lasting healing action, stress-controlled release, and regeneration of aged asphalt. As demonstrated in recent studies, asphalt mixtures containing these capsules have superior self-healing behavior in the fracture energy recovery test as the fracture healing energy even reaches 180% [24]. Nevertheless, these conclusions are reached under ideal circumstances.

Under actual service conditions, asphalt pavements will be gradually aged. Once the asphalt pavement ages, the bonding ability of asphalt binder will be decreased. At this time, under the dual effects of temperature fatigue and load stress fatigue, the aged asphalt concrete is more prone to generate fatigue microcracks [25,26]. In the process of pavement construction, the short-term aging of asphalt is dominated by thermal-oxidative aging. Thermal oxygen aging makes the pavement hard and brittle due to the thermal condensation reaction between molecules and volatilization of light components at high temperatures [27,28]. In the service process, the long-term aging of asphalt is mainly caused by photo-oxidative aging [29,30]. Long-term aging leads to an increase in the low-temperature stiffness of the pavement and a decrease in the damage strain, which makes it easy to form temperature shrinkage cracks and leads to pavement splitting. In summary, the aging of asphalt binder should be considered in simulating the self-healing performance of asphalt pavements containing capsules under actual service conditions.

It is worth noting that the current research only consider the influence of calcium alginate capsules on self-healing property of fresh asphalt mixtures without considering the ageing level of the mixtures containing capsules. The most important function of calcium alginate capsules is to rejuvenate the aged asphalt binder by supplementing the lost light components. Hence, the healing ability of asphalt mixtures containing calcium alginate capsules highly rely on the ageing level of asphalt binder in the mixtures.

To the best of our knowledge, there is no relevant literature concerning the effects of ageing level on the self-healing properties of asphalt mixtures containing calcium alginate capsules. In view of this, this work focuses on the effect of ageing degree on the healing property of asphalt concrete containing the capsules. In this study, alginate/attapulgite composite capsules were fabricated by the orifice-coagulation bath method and the principle of ion exchange. Secondly, a series of test were conducted to characterize the main properties of fresh prepared capsules and aged capsules extracted from asphalt concrete with short-term and long-term ageing treatments. Thirdly, the self-healing levels of asphalt mixtures beams containing alginate/attapulgite composite capsules after different ageing level treatment were evaluated through the three-point bending (3PB) and fatigue-healing test. Finally, the oil release ratios of capsules within asphalt mixtures with different ageing levels were characterized through Fourier Transform infrared spectroscopy (FTIR) test. The research methodology of this paper is shown in Figure 1.

## 2. Materials and Methods

### 2.1. Raw Materials

Virgin asphalt with a density of 1.034 g/cm^3^, a penetration of 68 (0.1 mm) at 25 °C and a softening point of 48.4 °C was obtained from Ezhou, Hubei. Basalt coarse aggregate, fine aggregate and filler were used to prepare the mixtures. The raw materials used to prepare the capsules including sodium alginate, calcium chloride, sunflower oil and Tween 80, were all purchased from Sinopharm Chemical Reagent Co. LTD, Shanghai, China. Attapulgite was produced from Xuyi, Jiangsu and it can increase the encapsulation rate of the sunflower oil and enhance the strength of the capsule shell material due to its unique chain-layered pore structure and large specific surface area. The rejuvenator used as core material was sunflower oil purchased from a local supermarket. The viscosity of the sunflower oil was 0.285 Pa s and the density was 0.935 g/cm^3^ at 20 °C [22,23].

### 2.2. Synthesis of Capsules

The preparation process of the capsules was shown in Figure 2. In the first step, a 1.875 wt% sodium alginate solution was prepared. Secondly, attapulgite powder was dispersed evenly in the sodium alginate (SA) solution system by high-speed shearing. Subsequently, sunflower oil and Tween 80 were added, and the volume ratio of sunflower oil to sodium alginate solution was 1:10 and Tween 80 accounted for 5% of the volume of sunflower oil. Then the mixture was emulsified by high-speed shearing for 10 min with a shear rate of 4000 r/min to form an oil-in-water emulsion. At the same time the 2.5 wt% calcium chloride solution was prepared, and then the emulsion was dropped into the CaCl_2_ solution. After complete reaction, the capsules were filtered out, cleaned with water, dried and set aside.

### 2.3. Characterization of Capsules

In this study, the mechanical strength of the capsules was measured by uniaxial compression test at room temperature using the Instron 5967 pressure tester (Instron, Norwood, USA) with a loading rate of 0.5 mm/min. As the force increases during loading, a yield point was formed on the stress-displacement curve when the capsule was destroyed, and the force corresponding to the yield point was recorded as the strength of the capsule. The surface morphology and internal structure of the capsules were observed by scanning electron microscopy (SEM) (Zeiss Gemini 300, Oberkochen, Germany). Firstly, the capsule was cut into semicircles with a blade, then the surface of the capsule was sprayed with gold, and finally the cross-section of the capsule was scanned with an electron microscope at a voltage of 5 kV. The thermal stability of the capsules was determined by thermogravimetric analysis (TGA) using the synchronous thermal analyzer (TA TGA55, New Castle, Delaware, USA). Moreover, the oil content in the capsules was quantified according to the thermogravimetric curves. The heating range was from 30 °C to 1000 °C and the heating rate was 10 °C/min. Nitrogen was injected as a protective gas throughout the heating process, with an inlet rate of 300 mL/min.

### 2.4. Preparation of Asphalt Concrete with Different Ageing Levels

#### 2.4.1. Preparation of Fresh Asphalt Concrete

In this paper, a dense graded asphalt concrete AC-13 with 70# virgin asphalt was used for both asphalt mixtures with and without capsules. The aggregate grade curve was shown in Figure 3. The aggregate was basalt and its main composition was showed in Table 1. The asphalt aggregate ratio was 4.8% and the capsule content was 0.5 wt% of the asphalt mixtures. Following the proportioning of the aggregates according to the target mix requirements, the aggregate was heated at 160 °C for 4 h. Then the aggregate was mixed with the asphalt at 160 °C in a mixing pot for 90 s. Next, the mineral powder was added and mixed for another 90 s. For fresh asphalt concrete in general, the mixture was poured into the mold to compact; for asphalt concrete containing capsules, the capsules were scattered on the surface of the mixtures and mixed for 90 s to sure the capsules mix well within the asphalt mixture. Finally, the loose asphalt mixtures (ordinary and containing capsules) were poured into a 300 mm × 300 mm × 50 mm steel mold and compacted using a rutting plate forming machine with a pressure of 900 kN, with 8 times of initial pressure and 28 times of recompression. After cooling at room temperature for 24 h, the rutting plates were cut into 95 mm × 45 mm × 50 mm beams with 4 mm × 10 mm notches directly underneath. The beams were used for the fracture-capsule release (healing)-re-fracture test.

#### 2.4.2. Preparation of Asphalt Concrete after Short-Term Ageing Treatment

Short-term aging of asphalt concrete was to simulate the aging of asphalt concrete from mixing to paving process. Firstly, the mixed mixtures were loosely laid in a tray at 21–22 kg/m^2^, then aged at 135 °C for 4 h (turning the mixtures every hour). Next, the mixtures were heated to 160 °C for rutting plate forming. Lastly, the rutting plates were cut into beams as described above.

#### 2.4.3. Preparation of Asphalt Concrete after Long-Term Ageing Treatment

Long-term aging of asphalt concrete was to simulate the ageing behavior of the asphalt pavement during the service time of 5–7 years. The experimental process was described as follows. Firstly, the specimens were placed into the oven under forced ventilation at 85 °C for 5 days. Then, the oven was turned off and the door was opened for natural cooling for 16 h. Lastly, the specimens were removed for further test.

### 2.5. Evaluation of Self-Healing Ratios of Asphalt Concrete with Different Ageing Levels after Different Cycles of Cyclic Loading

To simulate the self-healing performance of asphalt pavements in different ageing degrees under real vehicle loading, the fracture-cyclic compression loading-healing-refracture experiment was conducted using a universal test machine (UTM-25) from IPC Melbourne, Australia, with a maximum load of 25 kN and a temperature range of −20 °C to 60 °C. As shown in Figure 4, the fracture-cyclic compression loading-healing-refracture experiment included four steps: Step (1): three-point bending test was performed on asphalt concrete beams at −10 °C with a loading rate of 0.5 mm/min to simulate the growth of low-temperature cracks in asphalt concrete. The initial fracture strength of the beams before healing (F_1_) was obtained. Step (2): cyclic compression loading test was applied to the fractured beams in a steel mould to simulate the traffic loading of asphalt pavement at 20 °C according to previous research. The standard wheel pressure of 0.7 MPa and a frequency of 1 Hz were selected to conduct different cycles of compression loading on the asphalt concrete beams, simulating the wheel rolling of vehicles with different ages. Step (3): the asphalt concrete beams suffered the compression loading were healed at 20 °C for 48 h. Step (4): three-point bending test was repeated on the healed beams as described in Step (1) to obtain the fracture strength after healing (*F*_2_). The strength recovery ratio was defined as Equation (1)
(1)$$HI_{S}=\frac{F_{2}}{F_{1}}$$
where *F*_1_ was the initial fracture strength before healing, *F*_2_ was the fracture strength after healing and *HI*_S_ is the strength recovery ratio.

### 2.6. Quantitative Analysis of Asphalt Rejuvenator Release from Capsules

Nicolet 6700 Fourier Transform Infrared Spectrometer (Thermo Fisher Scientific, Waltham, MA, USA,) was employed to explore the release of rejuvenator in capsules to asphalt concrete beams after different cycles of compression loading. The asphalt mixture beam specimens were firstly heated in an oven at 70 °C for 30 min and then the beams were scattered by hand to pick out the capsules. The loose asphalt mixture without capsules was dissolved with trichloroethylene for 2 days and the upper liquid was extracted and placed in a fuming cupboard for 24 h to evaporate the trichloroethylene and obtain the extracted asphalt binders. Before FTIR testing, 0.1 g asphalt binder was added into the centrifuge tube and 2 mL carbon disulfide was dropped into tube to dissolve asphalt binder. Then the supernatant was dropped onto a KBr wafer and dried to form a layer of asphalt film on the surface of the wafer. The experimental parameters were set to scan the infrared spectrum of the sample in the range of 4000–400 cm^−1^ with a resolution of 4 cm^−1^ and a cumulative number of scans of 10 times. The oil release ratio was designed to characterize the amount of sunflower oil released in the bitumen from the capsule. It was defined as the percentage of the amount of oil released from the capsules to the total amount of oil wrapped in the capsules. According to the comparison of FIIR spectra, sunflower oil had a clear carbonyl absorbance peak of ester group at 1745 cm^−1^ while bitumen did not, see Figure 5. Asphalt had a strong absorbance peak of aromatic hydrocarbons carbon-carbon double bond at 1600 cm^−1^ while sunflower oil did not. Furthermore, the infrared spectrum of sunflower oil and asphalt have similar absorption peak in 3000–2750 cm^−1^ and fingerprint region (1300–400 cm^−1^). Since the peak area at 1600 cm^−1^ in the asphalt was almost constant relative to that of the full spectrum, the oil release rate could be calculated by comparing the peak area at 1745 cm^−1^ with that at 1600 cm^−1^, as displayed in Equation (2):(2)$$I_{1745{\mathrm{cm}}^{-1}}=\frac{\mathrm{Area}\;\mathrm{of}\;\mathrm{the}\;\mathrm{carbonyl}\;\mathrm{centered}\;\mathrm{around}\;1745{\;\mathrm{cm}}^{-1}}{\mathrm{Area}\;\mathrm{of}\;\mathrm{the}\;\mathrm{carbon}-\mathrm{carbon}\;\mathrm{double}\;\mathrm{bonds}\;\mathrm{of}\;\mathrm{aromatic}\;\mathrm{hydrocarbons}\;\mathrm{around}\;1600{\;\mathrm{cm}}^{-1}}$$

As shown in Figure 6, the standard relationship curve between the characteristic peak (1745 cm^−1^) area and the concentration of sunflower was obtained. Asphalt binder samples with oil contents of 0%, 1%, 2%, 4%, 6% and 8% were prepared by mixing the same sunflower oil and asphalt as used in the beams at 130 °C for 30 min. Then the linear relationship curve between sunflower oil content within asphalt and 1745 cm^−1^ peak area was fitted via FTIR results. The peak areas of the 1745 cm^−1^ of different extracted asphalt binders were calculated to acquire the oil contents in asphalt according to the oil content-peak area curve fitted.

## 3. Results and Discussion

### 3.1. Morphological and Interior Structure of Capsules after Different Ageing Treatments

Figure 7 showed the optical images of capsules in asphalt mixture with different ageing treatments and after different cycles of compression loading. Figure 7a–d presented the images of capsules during the fabrication process of fresh asphalt concrete and after 0 (after rolling compaction), 64,000 cycles of compression loading. It can be seen that the capsules after mixing and compaction still kept spherical shape similar to the original capsules, which indicated that the capsules could keep intact during the concrete fabrication process. It also can be seen that the capsules in fresh asphalt concrete after 64,000 cycles of compressive loading were slightly deformed but still kept intact. Figure 7e–g presented the images of capsules within asphalt mixtures with short-term ageing treatment, after 0, and 64,000 cycles of loading, respectively. The capsules in asphalt mixtures after short-term ageing treatment and compaction still kept complete shape, which implied that the capsules withstand the thermal and mechanical conditions in short-term ageing asphalt concrete. Furthermore, the capsules in asphalt concrete after 64,000 cycles of compression loading still deformed slightly and kept spherical shape. Figure 7h,i presented the images of capsules within asphalt mixtures with long-term ageingtreatment and after 64,000 cycles of compression loading, respectively. The capsules within asphalt concrete after long-term ageing treatment still deformed slightly and kept intact, which implied that the capsules presented good shape stability after the treatment (85 °C, 5 days). This indicated that the capsules in asphalt concrete can keep intact after several years of stimulated service period without compromising the volume performance of asphalt pavement.

Figure 8a1,a2 showed the SEM images of prepared calcium alginate/attapulgite capsules with multi-chamber structure and the healing agent was encapsulated in disjunctive chambers. The particular rejuvenator storage manner makes capsules own gradual oil release capacity and provides asphalt concrete containing capsules with long-term healing potential. It can be seen from the Figure 9b1,b3, Figure 10c1,c3 and Figure 11d1,d3 that the ageing treatment and cyclic loading will not damage the multi-chamber structure of calcium alginate/attapulgite capsules but will make the chamber wall become rough, which can be observed from Figure 9b2,b4, Figure 10c2,c4 and Figure 11d2,d4. The reason could be that the thermal treatment and compression loading make the capsules release encapsulated oil and lose the smoothness due to the leak-out of oil. Furthermore, the high temperature action may separate attapulgite from wall of the capsules.

In general, the capsules in asphalt concrete after different levels of ageing treatment and cyclic compression loading still kept intact shape similar to original capsules and the interior muti-chamber structure will not be damaged by the mechanical and thermal actions.

### 3.2. Mechanical Strength of Capsules after Different Ageing Treatments

Figure 12 showed the mechanical strength of capsules in asphalt mixtures with different levels of ageing treatment. The strength of capsules within fresh asphalt concrete after mixing, compaction and cyclic loading were presented in Figure 12a. It can be seen that the strength of capsules decreased from 15.9 N to 15.2 N and 14.7 N after mixing and compaction, respectively, which indicated that the capsules would lose little strength during the manufacturing process of asphalt mixture. The strength of capsules in asphalt mixture beam after 64,000 cycles of compressive loading was 13.4 N. This implied that the cyclic compression loading would decrease partly the strength of capsules, which was related to the release of the oil from the capsules.

Figure 12b showed the strength of capsules within asphalt mixtures after short-term ageing treatment, compaction and cyclic loading. It can be seen that the strength of capsules in asphalt mixtures after short-term ageing decreased obviously and decreased further after the cyclic compression loading. The strength of capsules within asphalt mixture beams after long-term ageing treatment and cyclic loading was showed in Figure 12c. The strength of capsules decreased apparently owing to the long-term ageing treatment compared with the original capsules.

In general, the original capsules meet the fabrication requirement of asphalt concrete in laboratory and the capsules lost little strength during the mixing and compaction process. The short-term ageing and long-term ageing treatment for asphalt mixtures would decrease the strength of the capsules. The long-term ageing made the strength of capsules decreased obviously. The strength of capsules in the three types of asphalt mixture beams all decreased gradually after the compression loading cycles.

### 3.3. Thermal Stability and Relative Oil Content of Capsules after Different Ageing Treatments

Figure 13a showed the mass loss of sunflower oil and capsules with and without oil. It can be seen that the sunflower oil began to volatilize at 306 °C and volatilize completely at 510 °C It can be also seen that the mass of capsules containing oil at 200 °C was less than 5%, which indicated that the capsules own good thermal stability and can resist the thermal condition during asphalt mixture fabrication. The mass of calcium alginate/attapulgite capsules decreased with the increase of heating temperature. From the room temperature to 300 °C, the mass of capsules released gradually due to the evaporation of free water and bound water in calcium alginate and the slight fracture of glycosidic protein in the chain structure of alginate. From 306 °C to 510 °C, the mass of capsules decreased rapidly. During this process, the encapsulated oil gradually volatilized completely and the glycoside bonds in the alginate chain structure were largely decomposed and decarbonized thus forming calcium carbonate and carbon dioxide. Furthermore, based on the mass loss of shell (capsule without oil), sunflower oil and capsules containing oil and according to the Equations (3) and (4), the relative oil content in capsules can be calculated. The oil content of calcium/attapulgite composite capsules was 54.9%.
(3)$$x+\left(y_{1}-y_{2}\right)\left(1-x\right)=z_{1}-z_{2}$$(4)$$x=\frac{z_{1}-z_{2}+y_{2}-y_{1}}{1+y_{2}-y_{1}}$$
where x is the relative oil content in the capsules (%), $y_{1\;}$ and $y_{2}$ are the residual mass percentage of capsule without oil at 306 ℃ and 510 ℃, respectively, and $z_{1}$ and $z_{2}$ are the residual mass percentage of capsules containing oil at 306 ℃ and 510 ℃, respectively.

Figure 13b–d presented the TGA curve of capsules in asphalt concrete after different levels of ageing and cyclic loading. It can be seen that the short-term ageing and long-term ageing treatments would not change the curve trend of capsules obviously compared with original capsules. Furthermore, comparing the curves of capsules in three type of asphalt concrete after 0 and 64,000 cycles of compression loading, the curves of 64,000 cycles were above the curves of 0 cycle, which indicated that the oil content of capsules after 64,000 cycles of loading was lower than that of capsules after 0 cycle (after compaction). It further implied that the capsules in fresh, short-term ageing and long-term ageing asphalt concrete can release its inner oil under cyclic compression loading.

### 3.4. Rejuvenator Release Ratio of Capsules within Asphalt Mixtures after Different Ageing Treatments under Cyclic Compression Loading

Figure 14 showed the oil release ratios (ORR) of capsules within asphalt mixtures during the preparation process. The ORR of capsules within fresh asphalt mixture after mixing before compaction was 5.9% and rose to 8.6% after compaction, which indicated that the ORR of capsules due to mixing was higher than that of capsules owing to compaction. The ORR of capsules within loose asphalt mixture with short-term ageing treatment was 31.8% after mixing before compaction and went up to 34.2% after compaction, which implied that the short-term ageing treatment (135 °C, 4 h) for asphalt mixture would make the capsules release about 25.9% of the oil inside. The released oil from capsules owing to short-term ageing may offset the adverse effect of ageing on asphalt properties. Furthermore, the ORR of capsules within asphalt concrete after short-term ageing treatment was 34.9% after compaction and went up to 43.4% after long-term ageing treatment, which implied that the long-term ageing treatment (85 °C, 5 days) for asphalt concrete would make capsules release about 8.5% of oil. In general, the capsules would release tiny amount of encapsulated oil prematurely during the asphalt mixture fabrication period and the released oil could decrease the thermal oxide ageing extent of asphalt during the mixing and compaction process. The short-term ageing and long-term ageing treatments in laboratory would increase the ORR of capsules and the capsules with short-term ageing treatment released more oil than capsules with long-term ageing treatment due to the high temperature action. The released oil may offset partly the adverse impact of ageing treatment on asphalt due to in-situ rejuvenation of sunflower oil.

The oil release ratios of capsules within asphalt mixture beams with different ageing treatments after different cycles of compression loading were presented in Figure 15. The oil release ratios of capsules in asphalt mixture beams without ageing treatment (WOA), after short-term ageing treatment and after long-term ageing treatment were 8.6%, 34.2% and 43.4%, respectively, corresponding to the change trend in Figure 14. The oil release ratios of capsules in fresh and aged asphalt concrete all increased with the increase of the cycles of compression loading, which implied that the capsules within asphalt concrete after different levels of ageing could release interior healing agent gradually with the cyclic loading. Furthermore, when the loading cycle was constant, the oil release ratio of capsules rose with the increase of the ageing level of asphalt concrete. For instance, the oil release ratios of capsules in asphalt concrete after slight thermal oxide ageing (fresh asphalt concrete), short-term ageing and long-term ageing treatment were 53.8%, 66.4% and 71.5% after 64,000 cycles of compression loading. The early ageing treatment for asphalt mixture would make capsules release some amount of oil before compression loading and thus increased the oil release ratio of capsules after fixed cycles of compression loading. It is noting that the oil release speed of capsules within fresh and aged asphalt concrete all decreased with the increase of loading cycles and the oil release speed of capsules in fresh asphalt concrete was higher than that of capsules in aged asphalt concrete. The reason could be that the stress-response ability of capsules would decrease with the loading cycles and the ageing treatment for asphalt concrete would make capsules become aged and thus slow down the oil release speed ulteriorly.

### 3.5. Self-Healing Ratios of Asphalt Mixture Beams with Different Ageing Levels after Cyclic Compression Loading

Figure 16 showed the bending strength recovery ratios of asphalt mixture beams without capsules with different levels of ageing treatment after cyclic compression loading. It is worth noting that the three types of asphalt concrete all regained partly strength without external loading after moderate healing period (20 °C, 48 h) due to the intrinsic healing capacity of asphalt binder. Moreover, the strength recovery ratio of asphalt mixture beams decreased with the increase of the ageing level of asphalt. The reason was be that due the ageing treatment asphalt binder became stiffer and thus reducing its flow ability.

The strength recovery ratios of fresh and aged asphalt mixture beams all slightly increased with the increase of compression loading cycles. The fractured asphalt mixture beams in steel mold were gradually compacted and the width of crack zone slightly decreased with the cycles of compression loading, thus slightly enhancing the strength recovery ratios of the three types of asphalt mixture beams without capsules. Furthermore, the strength recovery ratios of test beams decreased with the increase of the ageing level of asphalt mixtures after fixed compression loading cycles. For example, the strength recovery ratios of the three types of test beams were 41.2%, 39.3% and 34.2%, respectively, after 64,000 cycles of compression loading. The ageing treatments make the asphalt binder become stiffer and increase the viscosity, thus decreasing the capillary flow of asphalt in crack zone.

The strength recovery ratios of asphalt mixture beams containing capsules with different ageing levels after compression loading were presented in Figure 17 The strength recovery ratios of fresh, short-term ageing and long-term ageing beams containing capsules were 40.9%, 39.7% and 37.5%, respectively, without compression loading, which were higher than that of three types of asphalt mixture beams without capsules. The asphalt binder would become slightly aged during asphalt concrete fabrication, but the capsules would release some encapsulated oil owing to mixing, compaction and different levels of ageing treatment, thus rejuvenating the aged asphalt partly in-situ. Furthermore, the strength recovery ratio of fresh asphalt concrete was higher than that of aged asphalt concrete, which indicated that the released oil owing to ageing treatment failed to offset completely the negative effect of ageing.

The strength recovery ratios of the three types of test beams all increased with the increase of the loading cycles, which indicated that the fractured asphalt mixture beams with different levels of ageing treatment could regain strength recovery due to the sunflower oil released from capsules under the compression loading. Furthermore, when the loading cycles were constant, the strength recovery ratios of test beams decreased with the increase of the ageing level of asphalt mixture. For instance, the strength recovery ratios of the three types of asphalt mixture beams were 75.8%, 63.5% and 54.8%, respectively. The reason could be that the extra ageing treatment for asphalt concrete would make capsules become aged and thus slow down the oil release speed. Furthermore, the ageing treatment would make the released oil become aged and thus decrease the healing ability for aged asphalt. In general, the introduction of capsules into asphalt concrete with different levels of ageing treatment could obviously improve the strength recovery ratios of test beams under cyclic loading and the aggravation of ageing level would reduce the strength recovery ability of test beams without and with capsules.

## 4. Conclusions

In this study, calcium alginate/attapulgite composite capsules were prepared based on the reaction principle of ion exchange. The basic properties of prepared capsules and capsules extracted from fresh asphalt mixture and asphalt mixtures with short-term ageing and long-term ageing treatment were characterized, respectively. The self-healing levels of mixtures beams without and with capsules after different ageing treatments were evaluated. Meanwhile, the oil release ratios of capsules within asphalt mixtures containing capsules after different ageing treatments were characterized. Based on the experimental results, the following conclusion can be drawn:The prepared capsules showed muti-chamber structure. The capsules in asphalt concrete after different levels of ageing treatment and cyclic compression loading kept intact shape similar to original capsules and the interior structure will not be damaged by the mechanical and thermal actions.The original capsules meet the fabrication requirement of asphalt concrete in laboratory and the capsules lost little strength during the mixing and compaction process. The short-term ageing and long-term ageing treatment for asphalt mixtures would decrease the strength. The long-term ageing made the strengths of capsules reduced obviously. The strength of the capsules in the three types of asphalt mixture beams all decreased gradually after the cyclic compression loading.The capsules release less than 9% of the oil prematurely during the mixing and compaction process of asphalt mixture. Furthermore, the short-term ageing and long-term ageing treatment would induce the capsules released different amounts of oil in advance and the capsules after short-term ageing released about 26% of oil and the long-term ageing made capsules release about 8.5% of oil. The premature released oil in turn can offset the thermal oxide effect on asphalt partially.The oil release ratios of capsules within asphalt concrete with different levels of ageing increased with the compression loading cycles and the strength recovery of asphalt concrete after different ageing treatments also increased with the increase of the loading cycles owing to the healing effect of oil. The strength recovery of asphalt concrete decreased with the aggravation of asphalt ageing after the fixed loading cycles.Comparing the plain asphalt concrete, the introduction of calcium alginate capsules can prolong the service life of asphalt pavement owing to the healing capacity of asphalt rejuvenator released from the capsules.

This paper explored the healing properties of asphalt concrete with different levels of thermal oxidative ageing. It is worth noting that the simulative ageing treatments in laboratory are different from field ageing in real service condition of asphalt pavement, where temperatures are lower and the capsules will be less aged. However, compared with aged asphalt concrete without capsules, the strength recovery ratios of aged asphalt concrete containing capsules were much higher, which indicated that the introduction of calcium alginate/attapulgite capsules into aged asphalt concrete could obviously improve its healing properties. The ultraviolet ageing is along with asphalt pavement during the whole service period. Hence, the authors will focus on the healing properties of asphalt concrete containing capsules after UV ageing treatment in future research.

## Figures and Tables

**Figure 1 materials-15-01414-f001:**
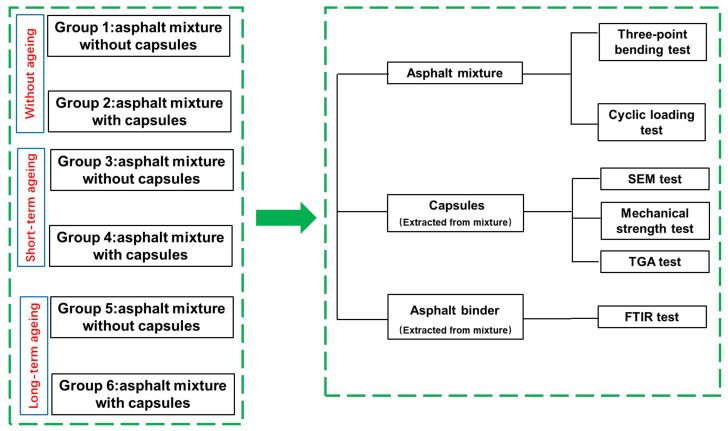
The research methodology of this research.

**Figure 2 materials-15-01414-f002:**
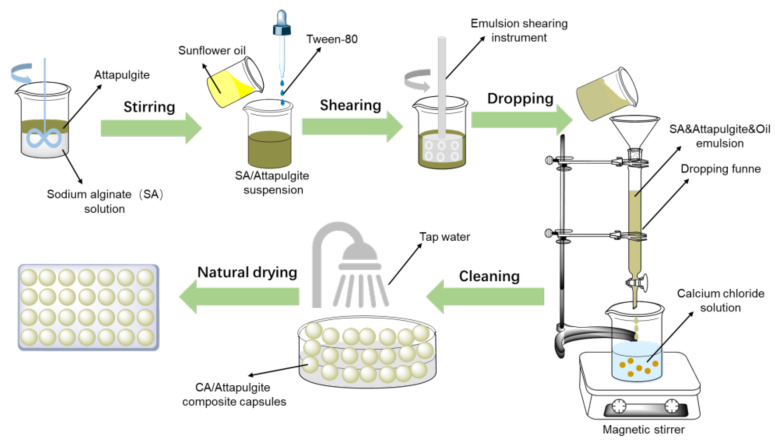
Preparation procedure of alginate/attapulgite composite capsules. Reproduced with permission from Ref. [31].

**Figure 3 materials-15-01414-f003:**
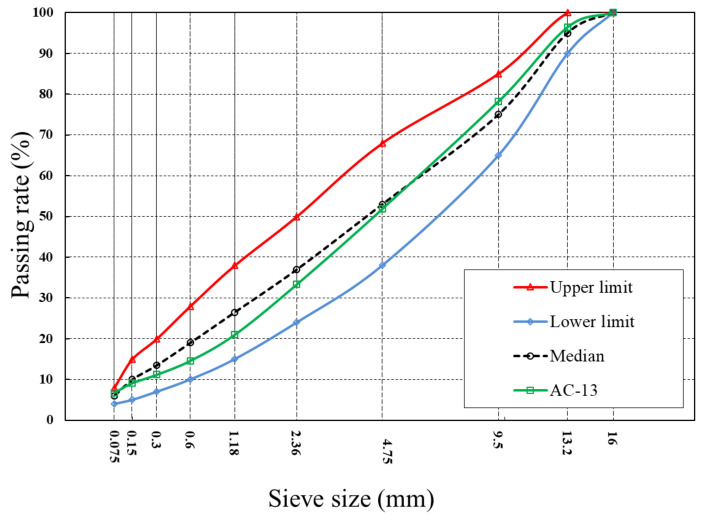
Aggregate gradation of the AC-13 asphalt mixture. Reproduced with permission from Ref. [31].

**Figure 4 materials-15-01414-f004:**
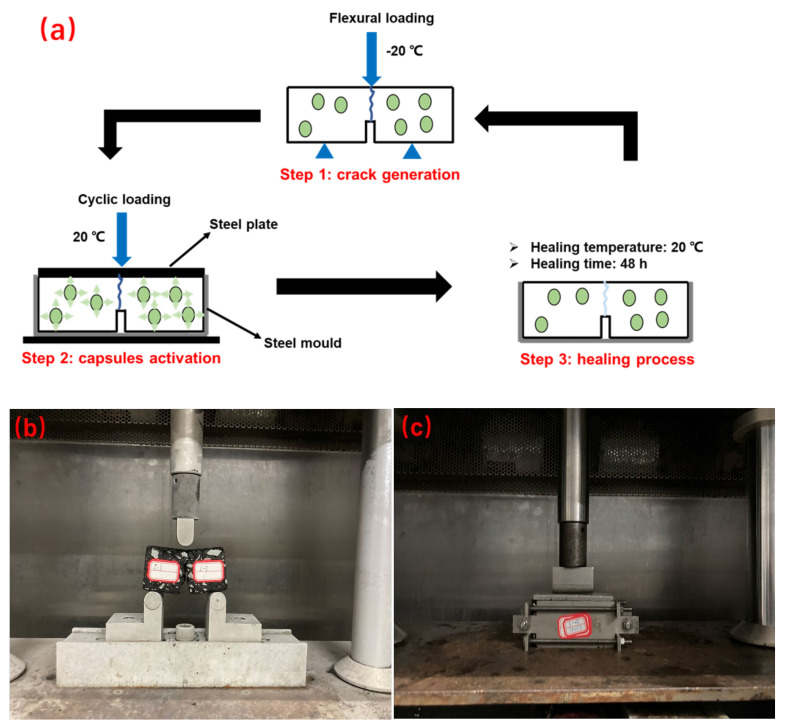
(**a**) Fracture-cyclic compression loading-healing-fracture test procedure, (**b**) 3PB test and (**c**) cyclic loading test.

**Figure 5 materials-15-01414-f005:**
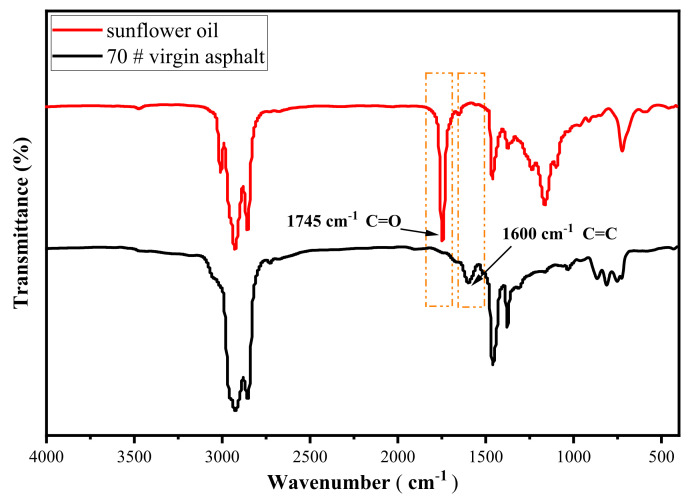
Infrared spectra of sunflower oil and virgin asphalt.

**Figure 6 materials-15-01414-f006:**
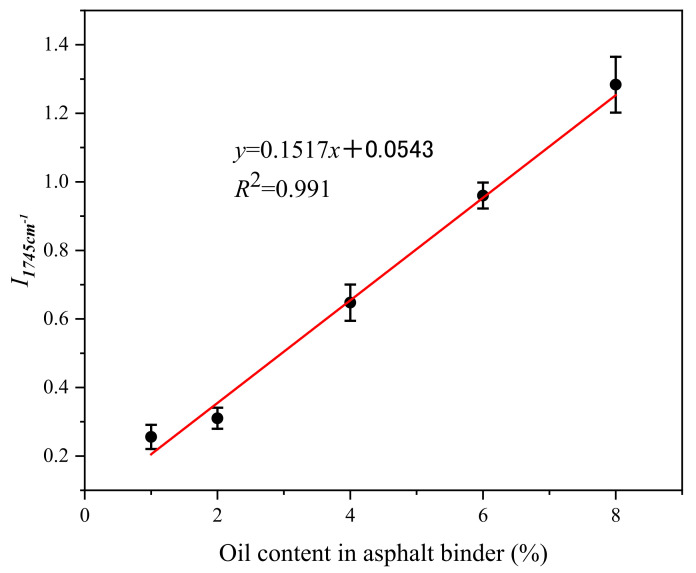
Standard relationship curve between $I_{1745\;{\mathrm{cm}}^{-1}}$ and oil content in asphalt binder.

**Figure 7 materials-15-01414-f007:**
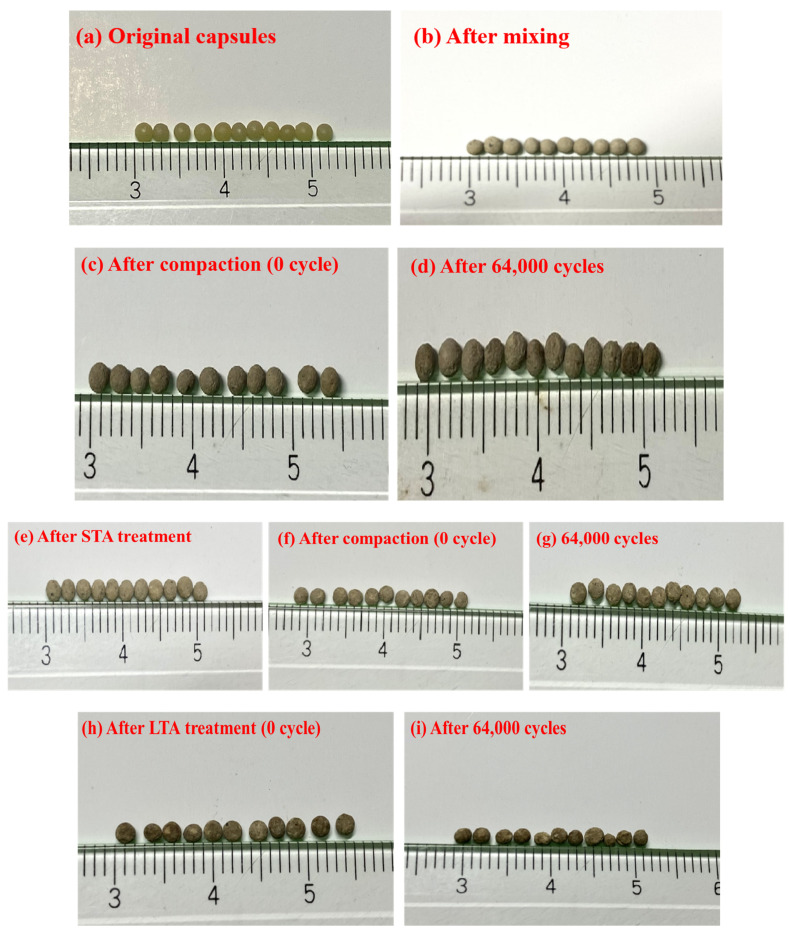
Images of capsules in asphalt mixture during concrete fabrication process and after different ageing treatments and loading cycles.

**Figure 8 materials-15-01414-f008:**
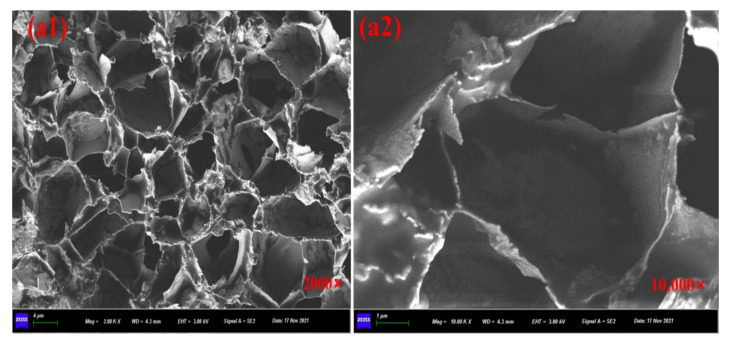
SEM images of original capsules. (**a1**) ×2000 and (**a2**) ×10,000.

**Figure 9 materials-15-01414-f009:**
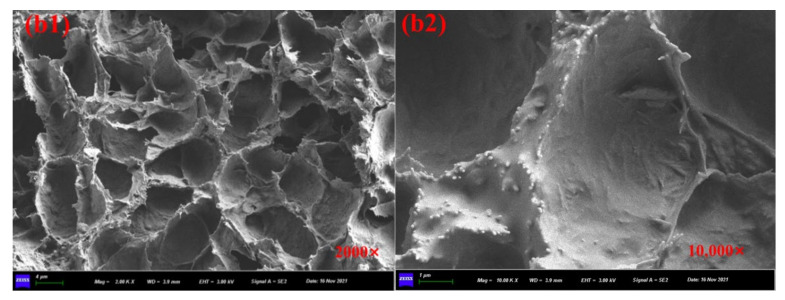
SEM images of the cross-section of capsules in asphalt mixtures. (**b1**,**b2**) capsules in asphalt mixture without ageing and after rolling compaction and (**b3**,**b4**) capsules in asphalt mixture without ageing and after 64,000 cycles of compression loading.

**Figure 10 materials-15-01414-f010:**
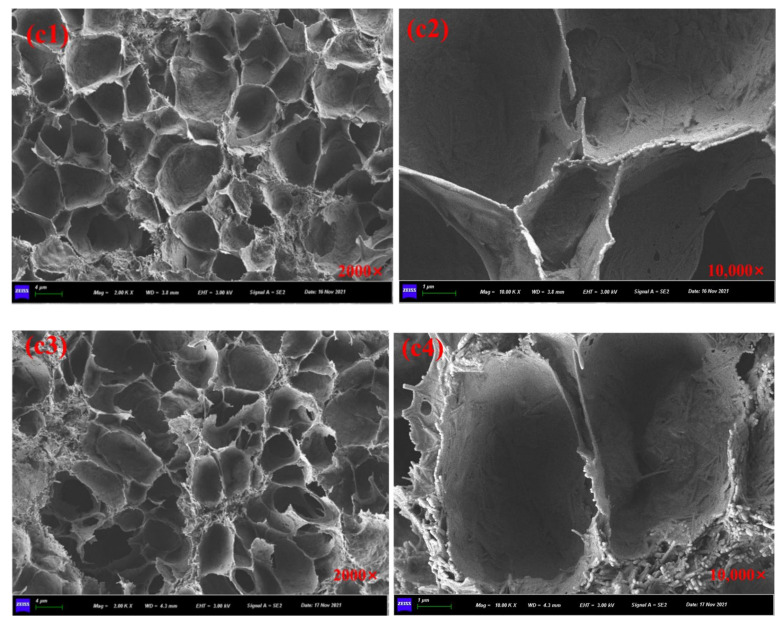
SEM images of the cross-section of capsules in asphalt mixtures. (**c1**,**c2**) capsules in asphalt mixture with short-term ageing treatment and after rolling compaction, (**c3**,**c4**) capsules in asphalt mixture with short-term ageing treatment and after 64,000 cycles of compression loading.

**Figure 11 materials-15-01414-f011:**
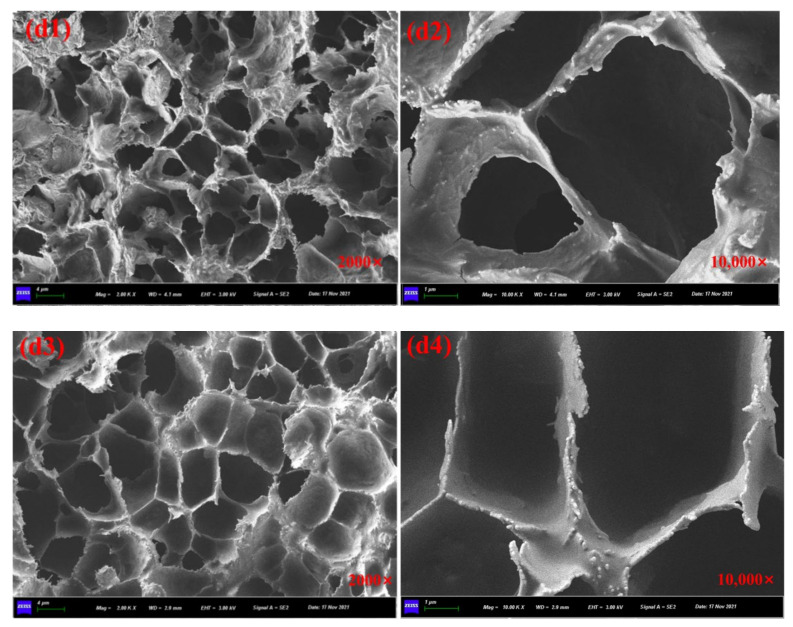
SEM images of the cross-section of capsules in asphalt mixtures. (**d1**,**d2**) capsules in asphalt mixture with long-term ageing treatment and after rolling compaction, (**d3**,**d4**) capsules in asphalt mixture with long-term ageing treatment and after 64,000 cycles of compression loading.

**Figure 12 materials-15-01414-f012:**
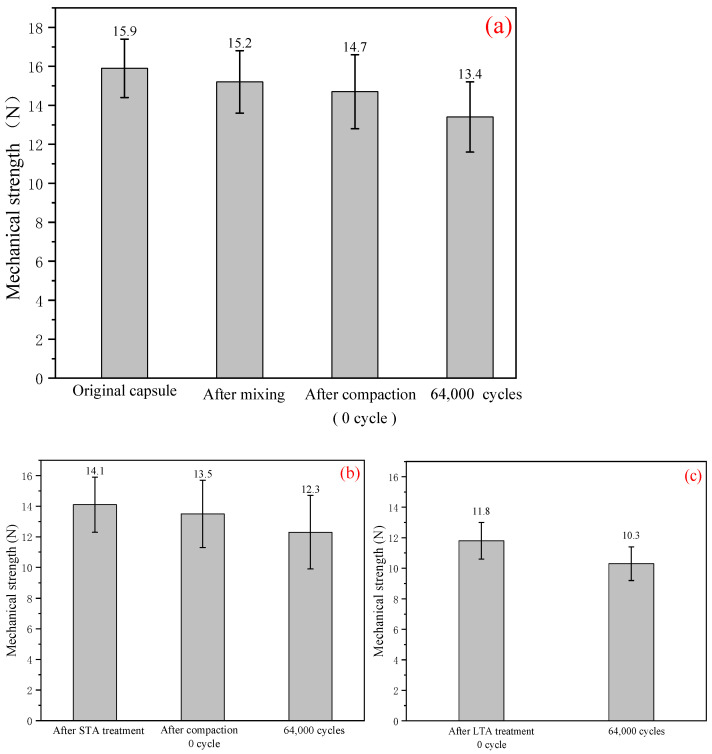
Mechanical strength of capsules within asphalt mixtures. (**a**) without ageing treatment, (**b**) with short-term ageing (STA) treatment and (**c**) with long-term ageing (LTA) treatment.

**Figure 13 materials-15-01414-f013:**
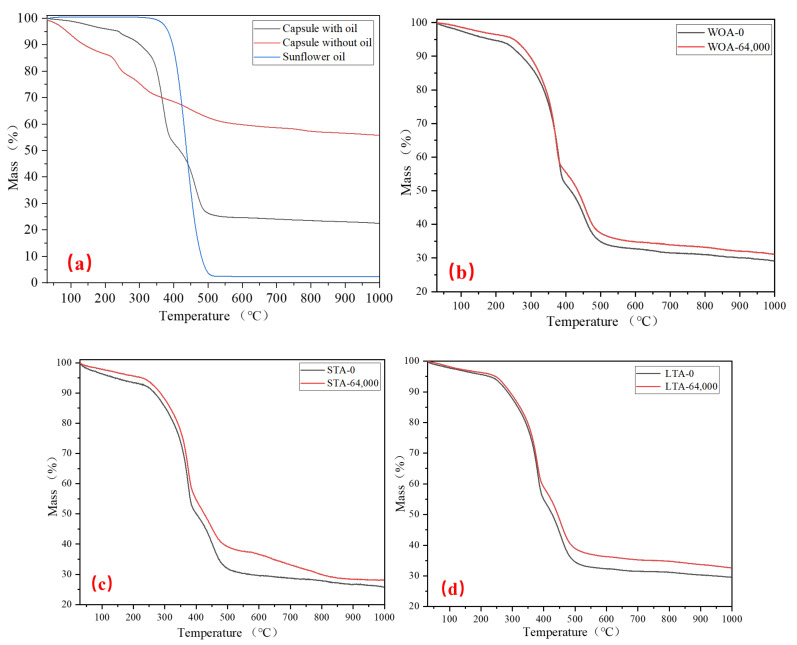
Thermal stability of capsules: (**a**) prepared capsules, (**b**) capsules in fresh asphalt concrete without ageing treatment (WOA) after different cycles of compression loading, (**c**) capsules in asphalt concrete with short-term ageing treatment after different cycles of compression loading and (**d**) capsules in asphalt concrete with long-term ageing treatment after different cycles of compression loading.

**Figure 14 materials-15-01414-f014:**
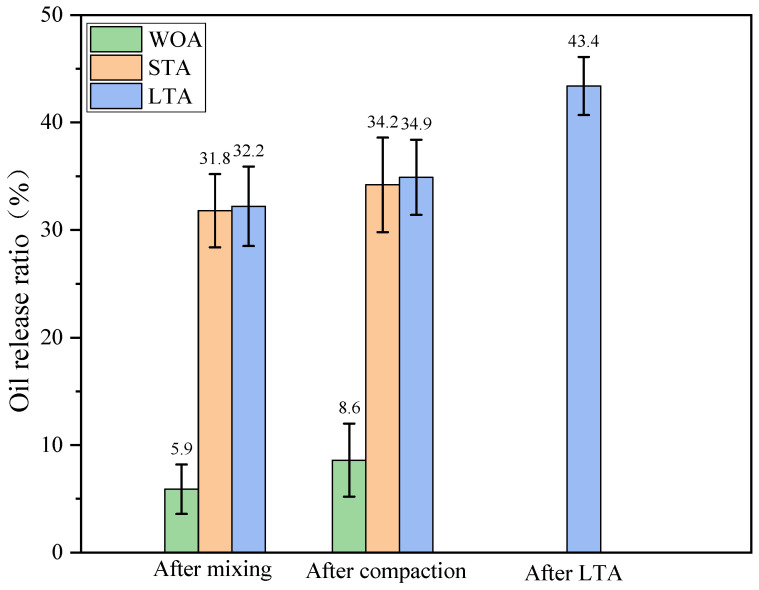
Oil release ratios of capsules within asphalt mixtures during the preparation process.

**Figure 15 materials-15-01414-f015:**
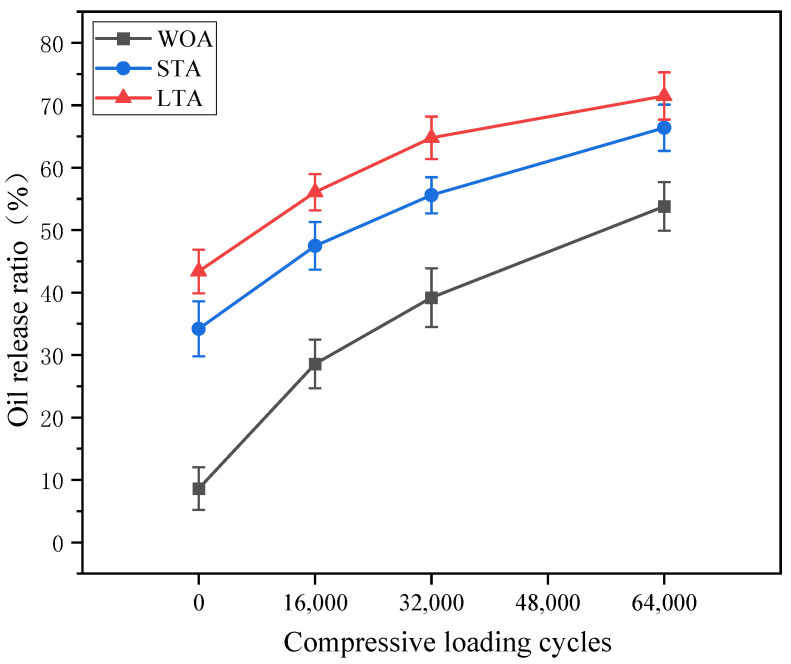
Oil release ratio of capsules within asphalt mixtures with different ageing treatments after cyclic compression loading.

**Figure 16 materials-15-01414-f016:**
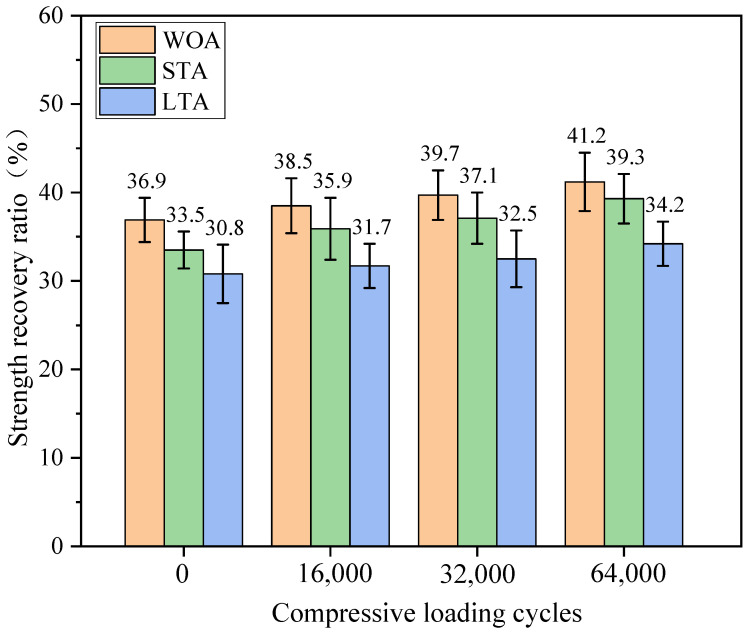
Strength recovery ratios of test beams without capsules with different ageing levels after cyclic compression loading.

**Figure 17 materials-15-01414-f017:**
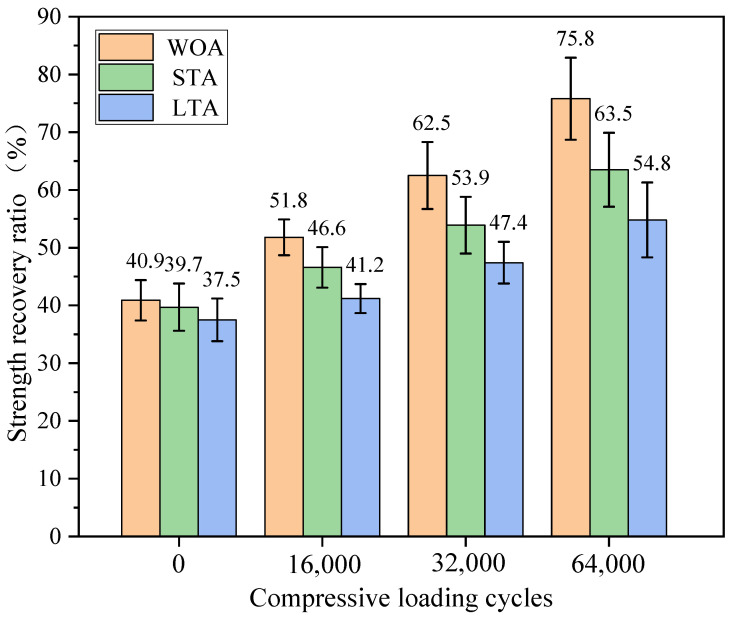
Strength recovery ratio of test beams with capsules with different ageing levels after compression loading.

**Table 1 materials-15-01414-t001:** The composition of basalt.

Compound	SiO_2_	Fe_2_O_3_	Al_2_O_3_	CaO	MgO	Na_2_O
Content	43.68%	13.66%	13.44%	12.18%	5.72%	2.52%

## Data Availability

Data is contained within the article.

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
