# Peer review of "Effect of Ageing on Self-Healing Properties of Asphalt Concrete Containing Calcium Alginate/Attapulgite Composite Capsules"

_materials, 2022, doi:10.3390/ma15041414_

Round 1

Reviewer 1 Report

The manuscript is devoted to the investigation of the mechanical behaviour and structural properties of composite capsules, containing calcium alginate attapulgite, on short and long term ageing period.
The paper provides a wide overview of the adoption of composite capsules for self-healing of asphalt mixture. The “material and methods” section provides a clear description of the followed procedures and of the adopted equipment. “Results and discussion” section clearly explains the results with high quality images (in particular, the SEM image) and graphs. Conclusions are synthetic and reports the main results of the study.
The paper is written with a very good English language and the reference section reports some recent literature works.

For all the previous reasons, the reviewer recommends accepting the paper in the present form for publication in Materials.

Author Response

Thank you very much for your positive comments.

Reviewer 2 Report

The topic of the paper is interesting and the paper is very well structured. The novelty of the paper is well stated and the question that authors tried to address in their research has been fully addressed. Therefore, I recommend accepting the paper in its current form.   

Author Response

(The authors gave the same response as above.)

Reviewer 3 Report

The manuscript focus on the effect of ageing on self-healing properties of asphalt concrete containing calcium alginate/attapulgite composite capsules. Generally, the methodology applied to solve this scientific problem is proper. Also the results are clearly presented. In my opinion, the manuscript is worth to investigate. However, I have some major comments before further processing:

  • Fig. 1 should be much more detailed to be readable itself without reading the supporting text – e.g. I would like to see more information about used capsules, I suggest to not use abbreviations there,
  • The list of references should be much more geographically diversified. Now most of the citations belongs to Chinese authors and according to my knowledge there is much progress in capsules worldwide,
  • General remark – I would rather prefer to see the full terms in the article rather than using permanently the abbreviations WOA, STA, LTA. This is hard to remember these abbreviations as they are not commonly used in the industry,
  • I have some suggestion regarding the performed literature survey. I suggest to not use citation pockets (e.g. [5-7]; [11-13] etc) but rather cite each reference individually (e.g. as investigated in [6]). If it is not possible I suggest to delete unnecessary reference,
  • Tab. 1 - Please provide a method how the properties of the sunflower oil were determined. If they are given based on manufacturers data I would like to see an information about it,
  • Section 2.4.1 – I think that it will be beneficial for the reader to provide the chemical composition of the aggregate together with the aggregate gradation,
  • Fig. 5 is not necessary and therefore I suggest to delete it from the article or merge it with fig. 4,
  • Fig. 7 - I suggest to not use statistical indicators (e.g. R) for the “correlation” based on just few measurements as it is statistically not significant. Additionally I would like to see error bars in this figure to see the scatter,
  • Fig. 13 – please describe on the figures what red dotted line means,
  • Authors contribution section has been missed in the article,
  • I would like to see more perspectives in conclusions section.

Reviewer 4 Report

I recommend the paper " Effect of ageing on self-healing properties of asphalt concrete containing calcium alginate/attapulgite composite capsules” for publication. Nevertheless, the paper required minor corrections and additions.

The Authors prepared calcium alginate/attapulgite composite capsules based on the reaction principle of ion exchange. The Authors perform laboratory tests that specified the basic properties of prepared capsules and capsules extracted from fresh asphalt mixture and asphalt mixtures with short-term ageing and long-term ageing treatment.

General report and comments:

  • 2. . Synthesis of capsules – The format of the section should be corrected (additional dot and space should be deleted).
  • Figure 3. Please increase font height. The blue lines are very similar. Please increase differences.
  • Chapter 3.5 (see Figures 17 and 18). The ANOVA statistical analysis should be performed to find out whether the results of laboratory tests obtained for defined groups of specimens differ significantly from each other.
  • In the concluding chapter, the future direction of investigation should be indicated according to reflect present investigation results.

Reviewer 5 Report

The manuscript entitled "Effect of ageing on self-healing properties of asphalt concrete containing calcium alginate/attapulgite composite capsules" presents an interesting experimental study conducted on the obtaining and characterization of asphalt concrete with self-healing capacity.  However, the results of the study have been presented with limited discussions and other issues must be addressed. The paper needs minor revisions before it is processed further, some comments follow:

Abstract

The abstract is written qualitatively. The majority of the qualitative statements should be modified for quantified result comparisons.

Materials and Methods Section

Figures 4 and 5 – Please introduce figure labels to highlight the areas of interest for the readers.

Figure 6 – please improve the description and discussions related to the FTIR spectra. There are peaks that haven’t been considered in the description.

Figure 8 – The capsules presentation is definitely valuable; however, a clear evaluation of the size evolution (average diameter) will increase the quality of the evaluation.

Figures 9 to 12 - Please introduce figure labels to highlight the areas of interest for the readers.

Round 2

Reviewer 3 Report

No more changes are required